# Application of the Sepsis-3 criteria to describe sepsis epidemiology in the Amsterdam UMCdb intensive care dataset

Christopher Y. K. Williams [1,2], Tom Edinburgh [1,3]*, Paul W. G. Elbers[4], Patrick J. Thoral[4], Ari Ercole[1,5]

**1** Department of Medicine, Division of Anaesthesia, University of Cambridge, Cambridge, United Kingdom, **2** Bakar Computational Health Sciences Institute, University of California, San Francisco, California, United States of America, **3** Department of Applied Mathematics and Theoretical Physics, University of Cambridge, Cambridge, United Kingdom, **4** Department of Intensive Care Medicine, Center for Critical Care Computational Intelligence, Amsterdam UMC, Vrije Universiteit, Amsterdam, The Netherlands, **5** Cambridge Centre for Artificial Intelligence in Medicine, University of Cambridge, Cambridge, United Kingdom

* te269@cam.ac.uk, thomas.edinburgh@hotmail.co.uk

**Data Availability Statement:** Data cannot be shared publicly because the underlying dataset is owned by a third-party (Amsterdam Medical Data

## Abstract

### Introduction

Sepsis is a major cause of morbidity and mortality worldwide. In the updated, 2016 Sepsis-3 criteria, sepsis is defined as life-threatening organ dysfunction caused by a dysregulated host response to infection, where organ dysfunction can be represented by an increase in the Sequential Organ Failure Assessment (SOFA) score of 2 points or more. We sought to apply the Sepsis-3 criteria to characterise the septic cohort in the Amsterdam University Medical Centres database (Amsterdam UMCdb).

### Methods

We examined adult intensive care unit (ICU) admissions in the Amsterdam UMCdb, which contains de-identified data for patients admitted to a mixed surgical-medical ICU at a tertiary academic medical centre in the Netherlands. We operationalised the Sepsis-3 criteria, defining organ dysfunction as an increase in the SOFA score of 2 points or more, while infection was defined as a new course of antibiotics or an escalation in antibiotic therapy, with at least one antibiotic given intravenously. Patients with sepsis were determined to be in septic shock if they additionally required the use of vasopressors and had a lactate level >2 mmol/L.

### Results

We identified 18,221 ICU admissions from 16,408 patients in our cohort. There were 6,312 unique sepsis episodes, of which 30.2% met the criteria for septic shock. A total of 4,911/6,312 sepsis (77.8%) episodes occurred on ICU admission. Forty-seven percent of emergency medical admissions and 36.7% of emergency surgical admissions were for sepsis. Overall, there was a 12.5% ICU mortality rate; patients with septic shock had a higher ICU mortality rate (38.4%) than those without shock (11.4%).

Science). This data is a large-scale de-identified database of intensive care unit patients who visited Amsterdam University Medical Centers between 2003 and 2016. The authors cannot share this data directly because access is subject to a formal application and data agreement with the data holders. The AmsterdamUMCdb dataset is freely accessible to researchers who have a) completed the required training course, b) signed the Access and End User License form, and c) submitted the prior forms for review. To gain access to the database, as described at https://www.amsterdammedicaldatascience.nl, the following steps are required: - Users must complete an appropriate training course for handling de-identified clinical data, such as Data or Specimens Only Research (DSOR) course from CITI (https://about.citiprogram.org/). - Users must submit a signed copy of the access agreement form. Once approved, users must create an account on EASY (https://easy.dans.knaw.nl/ui/home), complete their user profile and request download permission for the dataset on EASY. Once registered, users should then contact DANS (Data Archiving and Networked Services), who provide a link to the AmsterdamUMCdb archive. The authors did not receive any special privileges in accessing the data. Results and analysis are shared in a GitHub repository, as described at the end of the manuscript: https://github.com/tedinburgh/sepsis3-amsterdamumcdb.

**Funding:** TE is funded by Engineering and Physical Sciences Research Council (EPSRC) National Productivity Investment Fund (NPIF) EP/S515334/1, reference 2089662. The funders had no role in study design, data collection and analysis, decision to publish, or preparation of the manuscript.

**Competing interests:** The authors have declared that no competing interests exist.

**Abbreviations:** Amsterdam UMC, Amsterdam University Medical Center; HER, Electronic health record; HR, Hazard ratio; ICD, International classification of diseases; ICU, Intensive care unit; IQR, Interquartile range; LoS, Length of stay; MCU, Medium care unit; SOFA, Sequential organ failure assessment.

## Conclusions

We successfully operationalised the Sepsis-3 criteria to the Amsterdam UMCdb, allowing the characterization and comparison of sepsis epidemiology across different centres.

## Introduction

Sepsis is a major cause of morbidity and mortality worldwide, and is associated with significant hospital costs per patient [1,2]. Over the past few decades, there has been considerable focus on estimating the epidemiology of sepsis and, more recently, predicting the onset of sepsis using machine learning techniques [3–5]. However, these efforts are complicated by the absence of an accepted, objective diagnostic test for sepsis, alongside previously imprecise definitions (e.g *sepsis*, *severe sepsis*, and *septicaemia*) [6].

Consequently, an expert international Sepsis-3 task force was set up in 2016, through which consensus recommendations for the definition and clinical operationalisation of sepsis were established. These recommendations were termed the *Sepsis-3* criteria, being the third iterative update following 1991 and 2001 consensus recommendations [7,8]. In Sepsis-3, sepsis is defined as life-threatening organ dysfunction caused by a dysregulated host response to infection [9]. To operationalise this definition, the guidelines further state that organ dysfunction can be represented by an increase in the Sequential Organ Failure Assessment (SOFA) score of 2 points or more. One proposed definition for suspected infection is the sampling of microbial cultures and administration of antibiotics within a specific time period [9]. Patients with septic shock can be identified by a vasopressor requirement to maintain mean arterial pressure >65 mmHg and serum lactate levels >2 mmol/L in the absence of hypovolaemia. Crucially, the Sepsis-3 definitions were constructed to have a relationship to mortality, an essential prerequisite for face-validity.

This updated definition and recommendation for operationalising sepsis has led to studies beginning to apply the Sepsis-3 criteria to large electronic health records (EHRs) [10–12]. Defining sepsis using objective criteria, rather than relying on the more subjective International Classification of Diseases (ICD) coding, is thought to provide more reliable estimates of sepsis epidemiology [13]. The increasing availability of de-identified EHR datasets offers the opportunity to apply the Sepsis-3 criteria to intensive care unit (ICU) patients across a range of countries and patient cohorts, including in the United States and United Kingdom [10,12]. Since ICU population, structure and practice varies throughout the world, it is important that research datasets are properly characterised. We sought to apply the Sepsis-3 criteria to characterise the septic cohort in the Amsterdam University Medical Centres database, the first freely accessible European critical care database of complete ICU records [14], in order to contribute to the existing literature evaluating sepsis epidemiology in different settings.

## Methods

### Study population

The Amsterdam University Medical Centres database (Amsterdam UMCdb) is the first freely accessible European critical care database [14]. It contains de-identified data for 20,109 patients admitted to a mixed surgical-medical intensive care unit at a tertiary academic medical centre in the Netherlands, with up to 32 critical care beds and up to 12 high-dependency beds. Data was available from ICU admission to ICU discharge. Patients aged 18 years and

older admitted to the ICU were deemed eligible for inclusion. Exclusion criteria: patients admitted to the Medium Care Unit (MCU, high-dependency unit); patients with ICU admission duration of less than 1 hour; and patients with fewer than 3 Sequential Organ Failure Assessment (SOFA) component scores calculated on day 0 of ICU admission.

## Identification of sepsis and septic shock

In this study, we operationalised the identification of sepsis using the framework described by Shah et al in characterising the Critical Care Health Informatics Collaborative UK dataset [12]. Infection was defined as a new course of antibiotics (one or more doses of antibiotics prescribed for a patient not already receiving antibiotics) or an escalation in antibiotic therapy, with at least one antibiotic given intravenously. Following the antibiotic ranking classification of Braykov et al [15], an escalation in antibiotic therapy was defined as an increase in the maximum rank of current antibiotics from one 24 hour period to the next, or an increase in the number of antibiotics with the same maximum rank prescribed [12]. Organ dysfunction was defined as an increase in the Sequential Organ Failure Assessment (SOFA) score of 2 points or more [16]. Patients with sepsis were determined to be in septic shock if they additionally required the use of vasopressors and had a lactate level >2 mmol/L [9].

Full details, and accompanying code, of the operationalisation of sepsis in the Amsterdam UMC database can be found here [17,18]. For each patient admission in our dataset, daily SOFA scores (including both individual and total scores), daily antibiotic escalation status, and sepsis/septic shock status were calculated. Because pre-ICU data are limited in the Amsterdam UMC database, SOFA components were assumed to be zero for all patients prior to ICU admission as per the original Sepsis-3 recommendation. Patients with three or more missing SOFA components on day 0 of ICU admission were excluded. Among patients who died in the ICU, SOFA scores which could not be calculated on the day of death (due to missing SOFA components) were set to the maximum value of 24 [19]. A sepsis episode was identified when the SOFA score increased by $\geq 2$ on consecutive days with antibiotic escalation on either day, or when the SOFA score was at least 2 points higher on the day after antibiotic escalation compared to the day before [1]. Patients were eligible for a new (repeat) sepsis episode if the above criteria for sepsis was met more than 72 hours after a previous sepsis episode. Sepsis episodes occurring more than 15 days after ICU admission were excluded.

To accurately identify clinical suspicion of infection, prophylactic administration of antibiotics must be disregarded from the calculation of antibiotic escalation status. Notably, Amsterdam UMC ICU practises selective digestive decontamination including a four-day course of cefotaxime at ICU admission. In patients with suspected infection within this time period, the hospital practice is to exchange the cefotaxime prescription for the similar-spectrum antibiotic ceftriaxone. We therefore disregarded cefotaxime prescriptions within the first four days of ICU admission when calculating antibiotic escalation status. For cefotaxime prescriptions that extended beyond the first four days of ICU admission, we assumed that a suspected infection had occurred at some point within those first four days and that the cefotaxime prescription had (erroneously) not been exchanged for ceftriaxone. Due to uncertainty over when in this time period the prescription should have been changed, when calculating sepsis in these cases we allowed the timing of a SOFA increase $\geq 2$ to occur at any time on consecutive days between days 1 to 4 of ICU admission. Similarly, antibiotic use in the first 24 hours following ICU admission after elective surgery was assumed to be prophylactic and disregarded when calculating antibiotic escalation status. We additionally treated the following antibiotics as prophylactic (per local guidelines): vancomycin administration any day following cardiac surgery; all low dose (250 mg, 4 times daily) erythromycin administration; and all cefazoline administration.

## Statistical analysis

Mann-Whitney U and Kruskal-Wallis tests were used for statistical analysis of non-normally distributed continuous variables. The Chi-squared test was used to compare differences between categorical variables. We estimated age- and sex-adjusted hazard ratios of survival probability by admission sepsis status using cause-specific Cox proportional hazard models and Kaplan Meier curves. We carried out sensitivity analyses a) requiring >6 hours of noradrenaline infusion when calculating the cardiovascular component of SOFA score, b) where the criteria for suspected infection required both the prescription of one or more intravenous, non-prophylactic antibiotic and the presence of at least one microbial culture sample (from blood, urine, wound, catheter, faecal, drain, throat, nasal, rectal, or perineum cultures) within a 24 hour period, and c) where patients receiving a new or escalated antibiotic regimen, with associated SOFA score increase, for a single day only (as may occur for sick patients given empiric antibiotics but who are quickly identified as having non-infectious illness and are taken off antibiotics) were not classified as sepsis. $p < 0.05$ was considered statistically significant. Data were analysed using Python 3.9.12.

## Results

### Study characteristics

A total of 23,106 MCU/ICU admissions from the Amsterdam UMC database were examined in this study (Fig 1). Thirty-nine admissions were associated with an MCU/ICU stay duration < 1 hour and were excluded. Of the eligible admissions, 226 had fewer than three SOFA dimensions recorded in the first 24 hours and were excluded. This left 22,841 ICU/MCU admissions, of which 4,620 MCU admissions were excluded from the main analysis (S1–S3 Tables in S1 File). Our dataset therefore comprised 18,221 ICU admissions from 16,408 patients, including 7,397 (40.6%) elective surgical admissions, 1,741 (9.6%) emergency surgical admissions, and 9,083 (49.8%) medical admissions (Table 1 and S1 Fig). A majority of patients (66.1%) were over 60 years old on ICU admission, while 33.0% of patients were women. The median SOFA score was 6 (IQR 5–9) over the first 24 hours of admission. The most prevalent admitting specialty was Cardiothoracic Surgery (40.4%), followed by Cardiology (7.1%), Neurosurgery (6.0%), Gastrointestinal Surgery (5.2%), and General Internal Medicine (5.1%).

### Identification of sepsis and septic shock

Overall, we identified 33,025 occasions where SOFA score increased by ≥2 points, representing clinical deterioration, of which 6,592 (20.0%) were associated with antibiotic escalation and therefore classified as sepsis. After excluding 280 repeat sepsis classifications within 72 hours of first sepsis designation, this left 6,312 unique sepsis episodes. Among these, 2,032 (30.2%) sepsis episodes were associated with vasopressor use and lactate greater than 2 mmol/L, meeting the criteria for septic shock. There were 451 ICU admissions with >1 separate sepsis episode (424 admissions had two sepsis episodes, 27 admissions had >2 episodes), while 46 admissions had a second episode of septic shock >72 hours after the previous one.

A total of 4,911/6,312 sepsis episodes (77.8%) occurred on ICU admission, with 33.4% of these classified as septic shock on ICU admission. In a sensitivity analysis requiring a minimum of 6 hours for a noradrenaline infusion to be considered in the calculation of the cardiovascular SOFA score, there were 100 fewer emergency ICU admissions classified as septic shock compared to the main analysis, a difference that was statistically significant ($\chi2 = 10.0$, $p = 0.019$; S4 Table in S1 File). In a second sensitivity analysis where the criteria for suspected

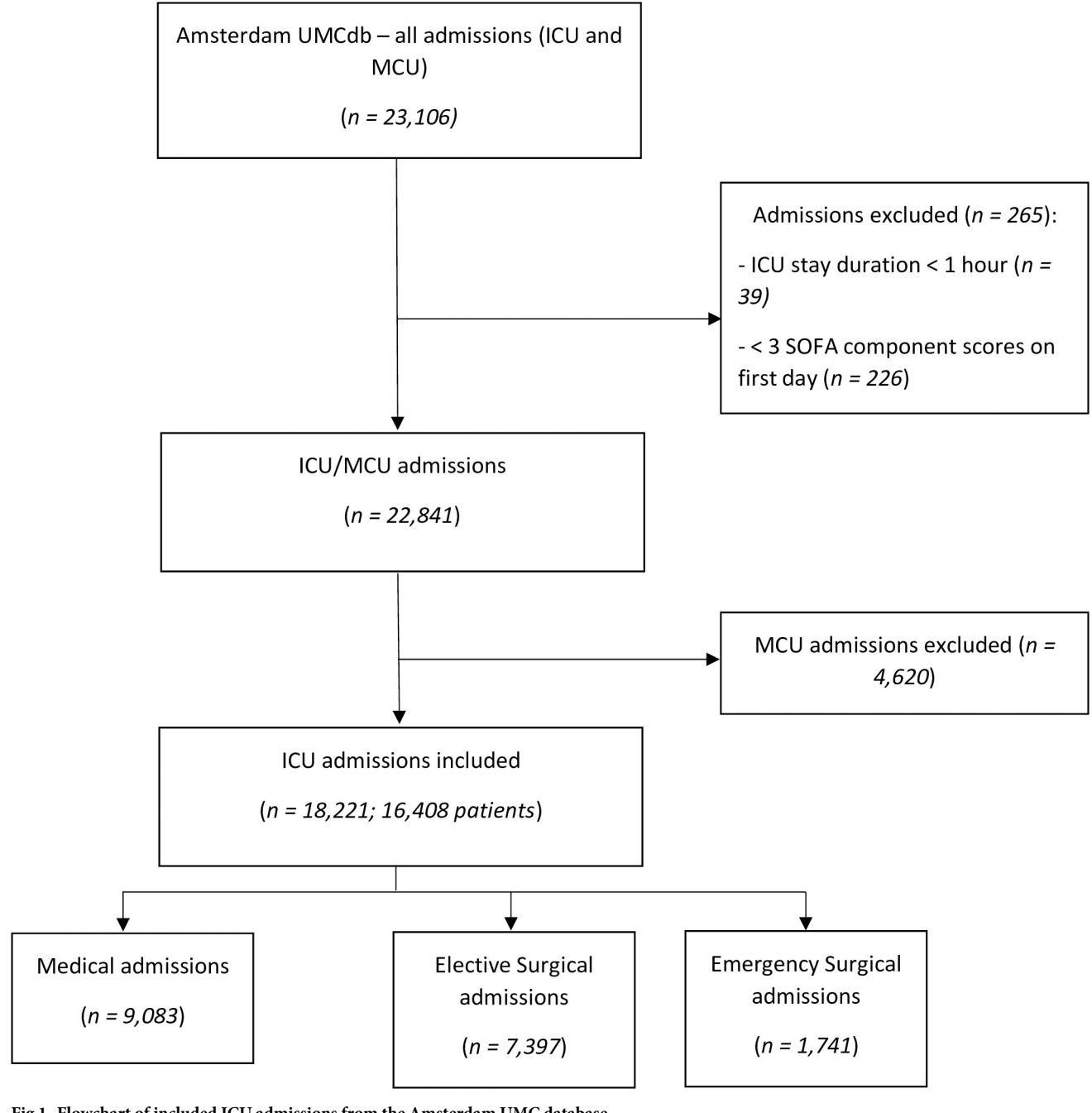

**Fig 1. Flowchart of included ICU admissions from the Amsterdam UMC database.**

infection required both microbial cultures to be taken and the prescription of intravenous, non-prophylactic antibiotics, there were significantly fewer emergency ICU admissions with sepsis (11.4% and 8.6% of all emergency ICU admissions had sepsis without shock and with shock, respectively, compared to 30.2% and 15.2% in the main analysis; χ2 = 2193.2, $p<0.001$; S5 Table in S1 File). Our third sensitivity analysis excluded patients who received a new or escalated antibiotic regimen, with associated SOFA score increase, for only a single day and led to 160 fewer emergency ICU admissions classified as either septic shock or sepsis without

**Table 1. Summary characteristics of ICU admissions by infection status.**

| | Overall | Septic shock | Sepsis without shock | Antibiotics without sepsis | Not on antibiotics |
|---|---|---|---|---|---|
| Number of admissions | 18221 | 1641 | 3270 | 4958 | 8352 |
| Number of patients | 16408 | 1612 | 3011 | 4709 | 8135 |
| Female, n (%) | 5876 (33.0) | 602 (37.9) | 1219 (38.8) | 1757 (35.5) | 2298 (28.3) |
| **Age group, n (%)** | | | | | |
| 18–39 | 1665 (9.1) | 203 (12.4) | 416 (12.7) | 406 (8.2) | 640 (7.7) |
| 40–49 | 1487 (8.2) | 176 (10.7) | 294 (9.0) | 428 (8.6) | 589 (7.1) |
| 50–59 | 3034 (16.7) | 246 (15.0) | 534 (16.3) | 819 (16.5) | 1435 (17.2) |
| 60–69 | 5018 (27.5) | 390 (23.8) | 822 (25.1) | 1288 (26.0) | 2518 (30.1) |
| 70–79 | 5157 (28.3) | 406 (24.7) | 857 (26.2) | 1425 (28.7) | 2469 (29.6) |
| 80+ | 1860 (10.2) | 220 (13.4) | 347 (10.6) | 592 (11.9+) | 701 (8.4) |
| **Admission category, n (%)** | | | | | |
| Elective surgical | 7397 (40.6) | 0 (0.0) | 0 (0.0) | 2866 (57.8) | 4531 (54.3) |
| Emergency surgical | 1741 (9.6) | 305 (18.6) | 334 (10.2) | 558 (11.3) | 544 (6.5) |
| Emergency medical | 9083 (49.8) | 1336 (81.4) | 2936 (89.8) | 1534 (30.9) | 3277 (39.2) |
| **Specialty, n (%)** | | | | | |
| Cardiothoracic surgery | 7369 (40.4) | 106 (6.5) | 615 (18.8) | 1832 (37.0) | 4816 (57.7) |
| Cardiology | 1300 (7.1) | 203 (12.4) | 133 (4.1) | 738 (14.9) | 226 (2.7) |
| Neurosurgery | 1089 (6.0) | 62 (3.8) | 129 (3.9) | 460 (9.3) | 438 (5.2) |
| Gastrointestinal surgery | 942 (5.2) | 164 (10.0) | 213 (6.5) | 317 (6.4) | 248 (3.0) |
| General Internal Medicine | 921 (5.1) | 216 (13.2) | 349 (10.7) | 135 (2.7) | 221 (2.6) |
| Vascular surgery | 897 (4.9) | 72 (4.4) | 109 (3.3) | 237 (4.8) | 479 (5.7) |
| Trauma | 774 (4.2) | 108 (6.6) | 203 (6.2) | 221 (4.5) | 242 (2.9) |
| Lung disease/surgery | 637 (3.5) | 77 (4.7) | 227 (6.9) | 179 (3.6) | 154 (1.8) |
| Neurology | 568 (3.1) | 57 (3.5) | 141 (4.3) | 179 (3.6) | 191 (2.3) |
| Haematology | 233 (1.3) | 63 (3.8) | 143 (4.4) | 9 (0.2) | 18 (0.2) |
| Gynaecology | 133 (0.7) | 20 (1.2) | 38 (1.2) | 27 (0.5) | 48 (0.6) |
| Urology | 132 (0.7) | 24 (1.5) | 31 (0.9) | 49 (1.0) | 28 (0.3) |
| Other medical specialty | 1286 (7.1) | 140 (8.5) | 431 (13.2) | 97 (2.0) | 618 (7.4) |
| Other surgical specialty | 331 (1.8) | 22 (1.3) | 46 (1.4) | 120 (2.4) | 143 (1.7) |
| From other ICU | 1121 (6.2) | 231 (14.1) | 328 (10.0) | 238 (4.8) | 324 (3.9) |
| **First 24hr physiology, median (IQR)** | | | | | |
| Maximum heart rate | 102 (89–119) | 123 (105–141) | 109 (93–127) | 101 (88–119) | 97 (87–110) |
| Minimum mean arterial pressure, mmHg | 56 (46–63) | 47 (36–55) | 55 (47–63) | 54 (43–62) | 58 (50–65) |
| Maximum FiO2 | 0.50 (0.41–0.61) | 0.70 (0.51–0.91) | 0.51 (0.41–0.80) | 0.50 (0.41–0.62) | 0.46 (0.40–0.55) |
| Minimum SpO2 | 0.94 (0.73–0.97) | 0.88 (0.72–0.95) | 0.95 (0.90–0.97) | 0.93 (0.70–0.97) | 0.95 (0.71–0.97) |
| Minimum PaO2, mmHg | 79 (68–95) | 71 (61–84) | 77 (66–94) | 78 (66–93) | 82 (71–99) |
| Minimum PaO2:FiO2 ratio | 195 (135–263) | 132 (88–198) | 180 (118–248) | 185 (128–253) | 215 (162–283) |
| Minimum GCS | 15 (10–15) | 11.0 (3.0–15.0) | 13.5 (7.0–15.0) | 15.0 (6.0–15.0) | 15 (14–15) |
| Maximum creatinine, μmol/L | 90 (72–119) | 132 (94–205) | 93 (71–125) | 87 (69–120) | 87 (72–106) |
| Minimum platelets | 150 (105–211) | 144 (78–216) | 163 (101–250) | 154 (108–215) | 146 (108–195) |
| Maximum bilirubin, μmol/L | 10 (6–17) | 14 (8–26) | 10 (6–17) | 10 (7–16) | 9 (6–14) |
| Maximum SOFA score | 6 (5–9) | 11 (8–13) | 7 (5–9) | 7 (5–10) | 5 (4–7) |
| Use of vasopressors, n (%) | 12254 (67.3) | 1641 (100.0) | 1708 (52.2) | 3711 (74.8) | 5194 (62.2) |
| Mechanical ventilation, n (%) | 15973 (87.7) | 1512 (92.1) | 2661 (81.4) | 4592 (92.6) | 7208 (86.3) |
| **Outcomes** | | | | | |
| Antibiotic escalation, first 24hr, n (%) | 5053 (27.7) | 1641 (100.0) | 3270 (100.0) | 142 (2.9) | 0 (0.0) |
| IV antibiotics for at least 4d (*), n (%) | 4438 (24.4) | 1443 (87.9) | 2909 (89.0) | 86 (1.7) | 0 (0.0) |

(*Continued*)

**Table 1.** (Continued)

|  | Overall | Septic shock | Sepsis without shock | Antibiotics without sepsis | Not on antibiotics |
|---|---|---|---|---|---|
| ICU length of stay, h, median (IQR) | 31 (22–114) | 140 (49–339) | 66 (24–203) | 62 (24–177) | 23 (20–41) |
| ICU mortality, n (%) | 2270 (12.5) | 630 (38.4) | 373 (11.4) | 816 (16.5) | 451 (5.4) |

*IV (non-prophylactic) antibiotics for at least 4 days in total *or* until ICU death/discharge.

shock compared to the main analysis ($\chi 2$ = 8.72, *p* = 0.033 S6 Table in S1 File). Such a situation may occur among sick patients who get empiric antibiotics in the ICU but are quickly identified as having non-infectious illness and antibiotics are stopped.

When stratified by admission category, 47.0% (4,272/9,083) of medical admissions were for sepsis, of which 31.3% (1,336/4,272) progressed to septic shock (Table 1). This compares to 36.7% (639/1,741) of emergency surgical admissions for sepsis, of which 47.7% (305/639) progressed to septic shock (Table 1). Eighty-nine percent (2,909/3,270) of ICU admissions for sepsis without shock were associated with IV antibiotics for at least 4 days or until ICU death or discharge, compared to 87.9% (1,443/1,641) admissions with septic shock. ICU stays were longest among patients with sepsis, either without shock (median length of stay [LoS] 66 hours, interquartile range [IQR] 24–203 hours) or with septic shock (median LoS 140 hours, IQR 49–339 hours), compared to patients on antibiotics without sepsis (median LoS 62 hours, IQR 24–177 hours) and those not on antibiotics (median LoS 23 hours, IQR 20–41 hours) (*p* < 0.001).

## Evolution of SOFA scores

Patients with septic shock on admission to the ICU had the highest maximum SOFA score on average (median 11, IQR 8–13), in contrast to patients not on antibiotics who had the lowest SOFA scores (median 5, IQR 4–7) (Table 1). Fig 2 shows component SOFA scores in the days following an ICU sepsis episode. Overall, SOFA scores improved after the start of the sepsis episode in those who survived a sepsis episode until discharge from the ICU (Fig 3) and deteriorated in those with sepsis who died in the ICU (Fig 4).

## ICU mortality

Among all ICU admissions there was a 12.5% ICU mortality rate. Patients with septic shock had a higher ICU mortality rate (38.4%) than those without shock (11.4%). In addition, among all patients with sepsis (with or without shock), patients who received <4 days antibiotics (with an ICU length of stay ≥4 days) had a lower mortality rate compared to those who received antibiotics for at least 4 days or until death (24.2% compared to 35.8% mortality rate, respectively) (Table 2). The trajectories of patients, stratified by sepsis status on admission, are shown in Fig 5. In age- and sex- adjusted all-mortality Cox proportional hazard models, ICU admission with septic shock and sepsis without shock was associated with hazard ratios of 1.67 (95% CI, 1.50–1.83) and 0.69 (95% CI, 0.61–0.77) respectively compared to without sepsis (Fig 6A and S7 Table in S1 File). In addition, admission SOFA score was associated with a significantly increased risk of ICU mortality for all ICU admissions (Fig 6B), and for each sepsis subgroup except 'Sepsis without shock' (Fig 6C–6F and S8 Table in S1 File).

## Antibiotic use

Of the 18,221 ICU admissions, 11,008 never received an antibiotic (excluding prophylactic antibiotics), 2,989 received one antibiotic regimen and 4,224 received two or more antibiotic regimens. There were 13,650 antibiotic courses prescribed over 58,341 patient-days; 21.7% of

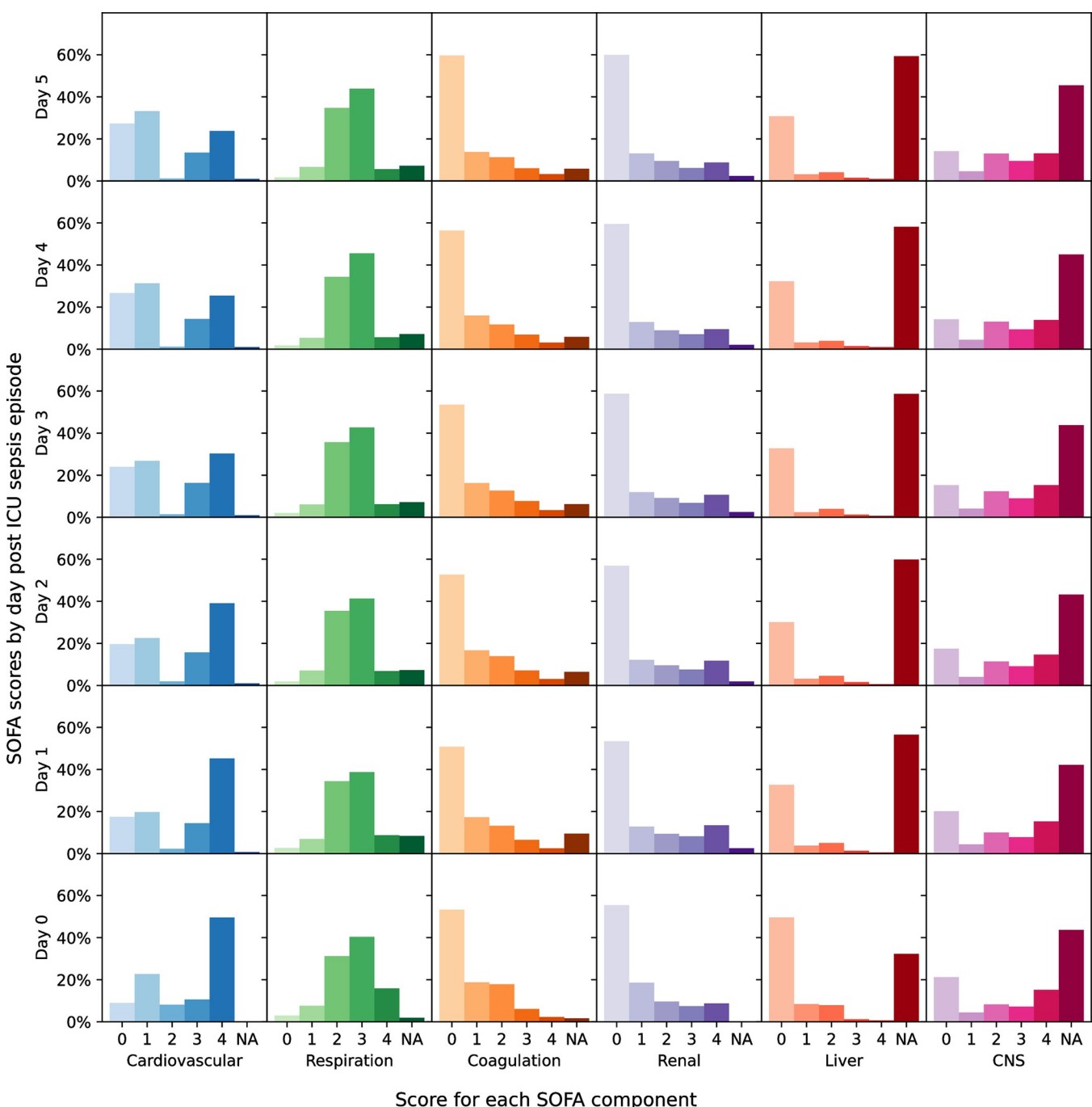

**Fig 2. Change in component SOFA score in the days following ICU sepsis episode.**

antibiotic courses were for narrow spectrum (rank 1) antibiotics, 51.6% broad spectrum (rank 2), 20.7% extended spectrum (rank 3) and 6.0% restricted (rank 4) (Table 3). The most common antibiotic prescriptions were ceftriaxone (27.1%), vancomycin (13.0%), metronidazole (9.7%), ciprofloxacin (7.7%) and co-amoxiclav (6.2%). The relative usage of treatment-only antibiotics for all patients in the ICU is shown in Fig 7.

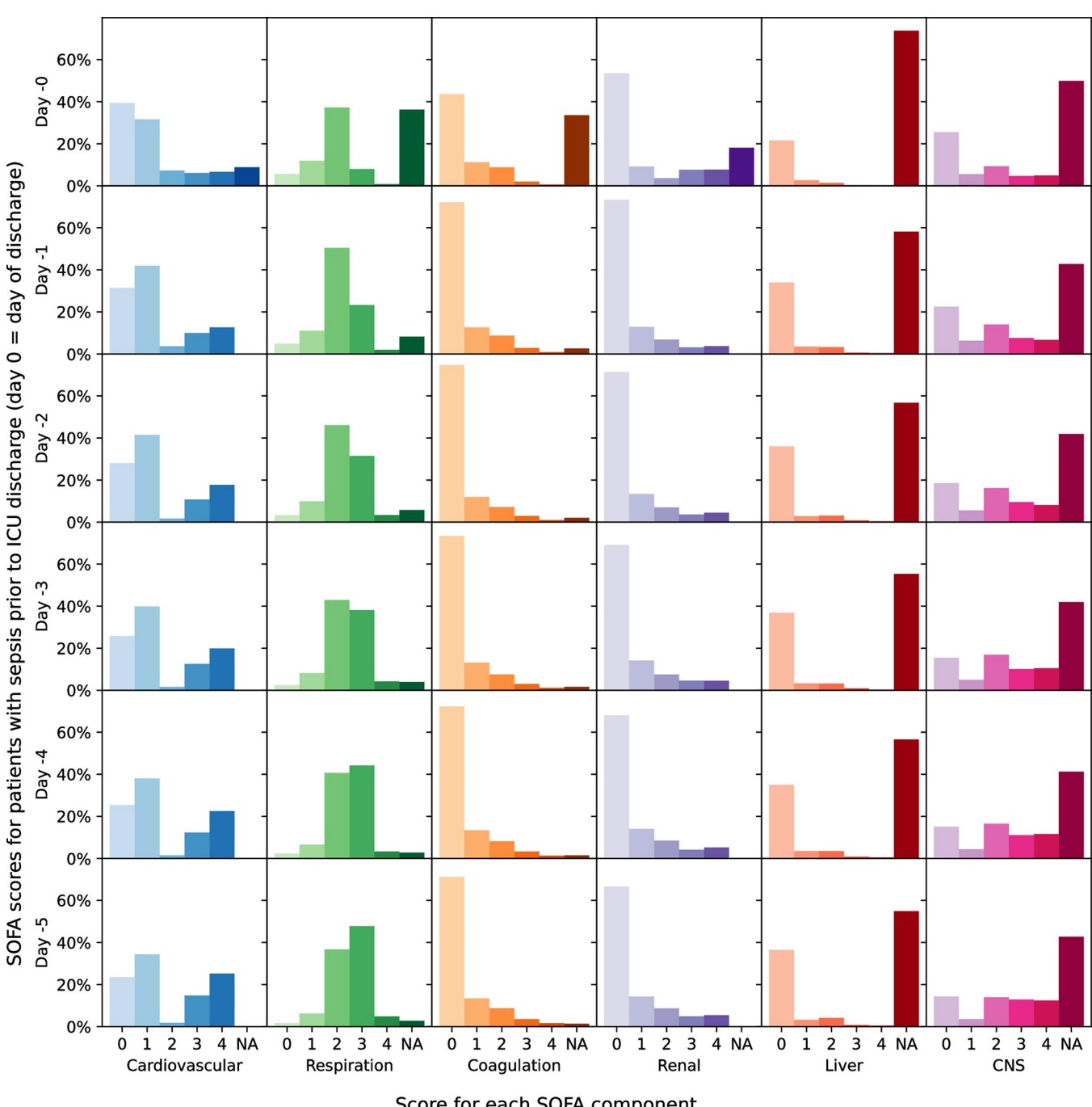

**Fig 3. Change in component SOFA score in the 5 days prior to ICU discharge among patients with sepsis (with and without shock).**

## Discussion

### Summary of findings

In this retrospective study of 18,221 admissions to the Amsterdam University Medical Center ICU, we found that 47.0% of emergency medical admissions and 36.7% of emergency surgical admissions had sepsis, of which 31.3% and 47.7% respectively progressed to septic shock. ICU

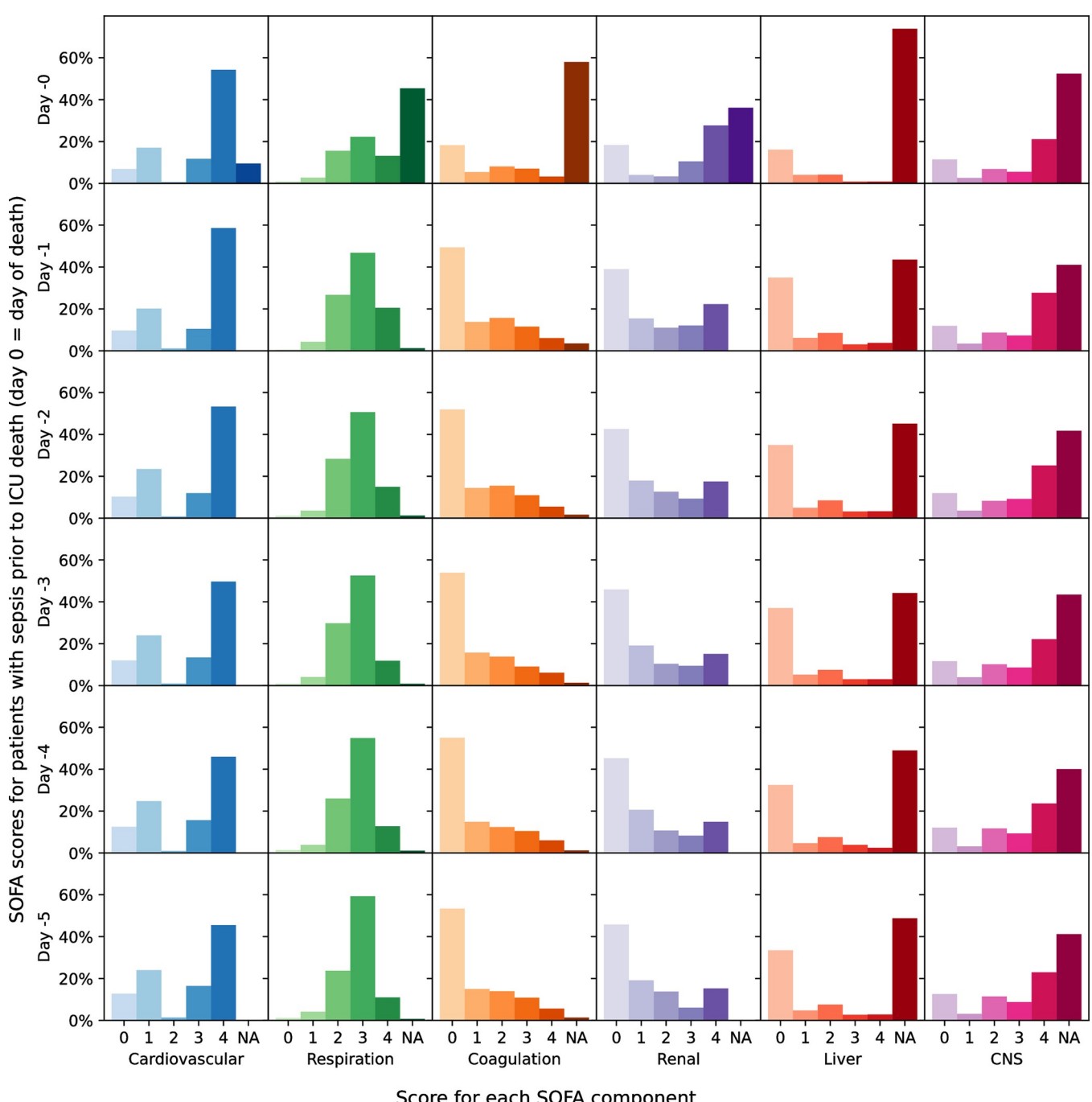

**Fig 4. Change in component SOFA score in the 5 days prior to ICU death among patients with sepsis (with and without shock) who died intra-ICU.**

admissions with sepsis but no shock had better survival (HR 0.69 [95% CI 0.61–0.77]) than admissions without sepsis, whereas patients with septic shock had significantly worse survival (HR 1.67 [95% CI 1.50–1.83]). Higher admission SOFA scores were associated with increased ICU mortality risk, with daily SOFA scores increasing in patients who died of sepsis and decreasing among those who survived to ICU discharge.

**Table 2. Summary characteristics of ICU admissions with sepsis (with or without shock), by duration of antibiotics.**

| | Antibiotics for at least 4 days or until death | Antibiotics for < 4 days | | Overall (all patients with sepsis on admission) |
| --- | --- | --- | --- | --- |
| | | Antibiotics until discharge (ICU length of stay <4 days) | Antibiotics <4 days (ICU length of stay ≥4 days) | |
| Number of admissions | 2423 | 1929 | 559 | 4911 |
| Number of patients | 2295 | 1867 | 548 | 4473 |
| Women, n (%) | 882 (37.4) | 714 (39.0) | 225 (41.7) | 1821 (38.5) |
| **Age group, n (%)** | | | | |
| 18–39 | 289 (11.9) | 263 (13.6) | 67 (12.0) | 619 (12.6) |
| 40–49 | 237 (9.8) | 186 (9.6) | 47 (8.4) | 470 (9.6) |
| 50–59 | 389 (16.1) | 303 (15.7) | 88 (15.7) | 780 (15.9) |
| 60–69 | 596 (24.6) | 488 (25.3) | 128 (22.9) | 1212 (24.7) |
| 70–79 | 618 (25.5) | 498 (25.8) | 147 (26.3) | 1263 (25.7) |
| 80+ | 294 (12.1) | 191 (9.9) | 82 (14.7) | 567 (11.5) |
| **Admission category, n (%)** | | | | |
| Elective surgical | 0 (0.0) | 0 (0.0) | 0 (0.0) | 0 (0.0) |
| Emergency surgical | 321 (13.2) | 266 (13.8) | 52 (9.3) | 639 (13.0) |
| Emergency medical | 2102 (86.8) | 1663 (86.2) | 507 (90.7) | 4272 (87.0) |
| **Specialty, n (%)** | | | | |
| Cardiothoracic surgery | 151 (6.2) | 507 (26.3) | 63 (11.3) | 721 (14.7) |
| Cardiology | 214 (8.8) | 61 (3.2) | 61 (10.9) | 336 (6.8) |
| Neurosurgery | 118 (4.9) | 49 (2.5) | 24 (4.3) | 191 (3.9) |
| Gastrointestinal surgery | 215 (8.9) | 138 (7.2) | 24 (4.3) | 377 (7.7) |
| General Internal Medicine | 335 (13.8) | 170 (8.8) | 60 (10.7) | 565 (11.5) |
| Vascular surgery | 96 (4.0) | 64 (3.3) | 21 (3.8) | 181 (3.7) |
| Trauma | 151 (6.2) | 122 (6.3) | 38 (6.8) | 311 (6.3) |
| Lung disease/surgery | 162 (6.7) | 110 (5.7) | 32 (5.7) | 304 (6.2) |
| Neurology | 103 (4.3) | 52 (2.7) | 43 (7.7) | 198 (4.0) |
| Haematology | 151 (6.2) | 46 (2.4) | 9 (1.6) | 206 (4.2) |
| Gynaecology | 11 (0.5) | 36 (1.9) | 11 (2.0) | 58 (1.2) |
| Urology | 24 (1.0) | 30 (1.6) | 1 (0.2) | 55 (1.1) |
| Other medical specialty | 224 (9.2) | 278 (14.4) | 69 (12.3) | 571 (11.6) |
| Other surgical specialty | 30 (1.2) | 37 (1.9) | 1 (0.2) | 68 (1.4) |
| From other ICU | 316 (13.0) | 164 (8.5) | 79 (14.1) | 559 (11.4) |
| **First 24hr physiology, median (IQR)** | | | | |
| Maximum heart rate | 120 (103–138) | 105 (90–123) | 111 (96–132) | 113 (96–132) |
| Minimum mean arterial pressure, mmHg | 48 (38–55) | 59 (52–66) | 51 (42–60) | 53 (43–61) |
| Maximum FiO2 | 0.70 (0.51–0.90) | 0.50 (0.40–0.60) | 0.60 (0.41–0.90) | 0.60 (0.41–0.90) |
| Minimum SpO2 | 0.92 (0.78–0.96) | 0.96 (0.91–0.98) | 0.93 (0.81–0.96) | 0.94 (0.83–0.97) |
| Minimum PaO2, mmHg | 70 (61–84) | 82 (69–101) | 75 (65–88) | 75 (64–91) |
| Minimum PaO2:FiO2 ratio | 127 (87–187) | 211 (151–281) | 165 (108–227) | 162 (104–234) |
| Minimum GCS | 11.0 (3.0–15.0) | 15 (10–15) | 10.0 (3.0–15.0) | 13.0 (6.0–15.0) |
| Maximum creatinine, µmol/L | 112 (78–181) | 94 (74–121) | 106 (75–174) | 102 (76–153) |
| Minimum platelets | 162 (86–249) | 148 (97–225) | 164 (99–253) | 156 (94–240) |
| Maximum bilirubin, µmol/L | 12 (7–22) | 10 (6–18) | 11 (6–19) | 11 (7–20) |
| Maximum SOFA score | 9 (7–12) | 6 (4–8) | 8 (6–11) | 8 (6–11) |
| Use of vasopressors, n (%) | 1974 (81.5) | 979 (50.8) | 396 (70.8) | 3349 (68.2) |
| Mechanical ventilation, n (%) | 2238 (92.4) | 1469 (76.2) | 466 (83.4) | 4173 (85.0) |
| **Outcomes** | | | | |

*(Continued)*

**Table 2.** (Continued)

| | Antibiotics for at least 4 days or until death | Antibiotics for < 4 days | | Overall (all patients with sepsis on admission) |
| --- | --- | --- | --- | --- |
| | | Antibiotics until discharge (ICU length of stay <4 days) | Antibiotics <4 days (ICU length of stay ≥4 days) | |
| ICU length of stay, h, median (IQR) | 227 (122–449) | 26 (22–48) | 100 (52–225) | 85 (28–253) |
| ICU mortality, n (%) | 868 (35.8) | 0 (0.0) | 135 (24.2) | 1003 (20.4) |

## Comparison to other studies

By applying the Sepsis-3 criteria to Amsterdam UMCdb we were able to assess the epidemiology of sepsis in a manner that is comparable to other studies in the UK and USA [10,12]. Whilst sepsis is currently defined as life-threatening organ dysfunction caused by a dysregulated host response to infection, with organ dysfunction represented by an increase in SOFA score of 2 points or more [9], there remains uncertainty between studies as to how best to define suspected infection. In a UK-based study of four National Health Service hospital trusts, Shah et al chose to define this as a new course of antibiotics or an escalation in antibiotic therapy [12]. In contrast, using the publicly available MIMIC-III ICU database, Johnson et al employed the definition used by the original study from which the Sepsis-3 criteria was established, where both culture of bodily fluids and administration of antibiotics were necessary to classify infection [9,10,20].

We used the former criteria of antibiotic escalation to signify infection (regardless of culture status) in our main analysis and found comparable levels of septic shock (15.2% of emergency admissions compared to 17.9%) and overall sepsis (45.4% of emergency admissions compared to 60.4%) on ICU admission as were reported by Shah and colleagues [12]. Similar to Shah et al, we also found that patients with septic shock had significantly worse survival, while those with sepsis but no shock had better survival than those with no sepsis. This latter finding may reflect the severity of non-sepsis-related diseases that warrant ICU admission. Alternatively, it is possible that, in patients admitted with sepsis but no shock, treatment is commenced sufficiently early in the disease trajectory to prevent further deterioration. Future studies should examine the effect of sepsis with and without shock on outcomes other than ICU mortality, including longer-term mortality, post-discharge co-morbidity status and overall quality of life scores.

In our sensitivity analysis requiring >6 hours of noradrenaline infusion when calculating the cardiovascular component of SOFA score, a minority of ICU admissions originally classified as having septic shock were downgraded. When the criteria for infection was modified to include both the administration of antibiotics and the sampling of bodily fluid for culture, there were significantly fewer episodes of sepsis (both with and without shock) identified: only 20.1% of emergency ICU admissions were classified as for sepsis overall (with or without shock) and 8.6% for septic shock. This compares to 49.1% of patients identified by Johnson et al who met the Sepsis-3 criteria (sepsis with or without shock) in the MIMIC-III dataset. One possible explanation for the difference in admission sepsis prevalence on using this second criteria for infection is that, due to the limited availability of pre-ICU data in the Amsterdam UMCdb, cultures may have already been taken prior to ICU admission and were consequently not repeated in-ICU at the time of antibiotic escalation. Alternatively, there may be differences in the frequency of sampling bodily fluid for culture between institutions and/or countries. It is important that this disparity between methods of determining infection status is considered when describing sepsis epidemiology across other datasets in the future.

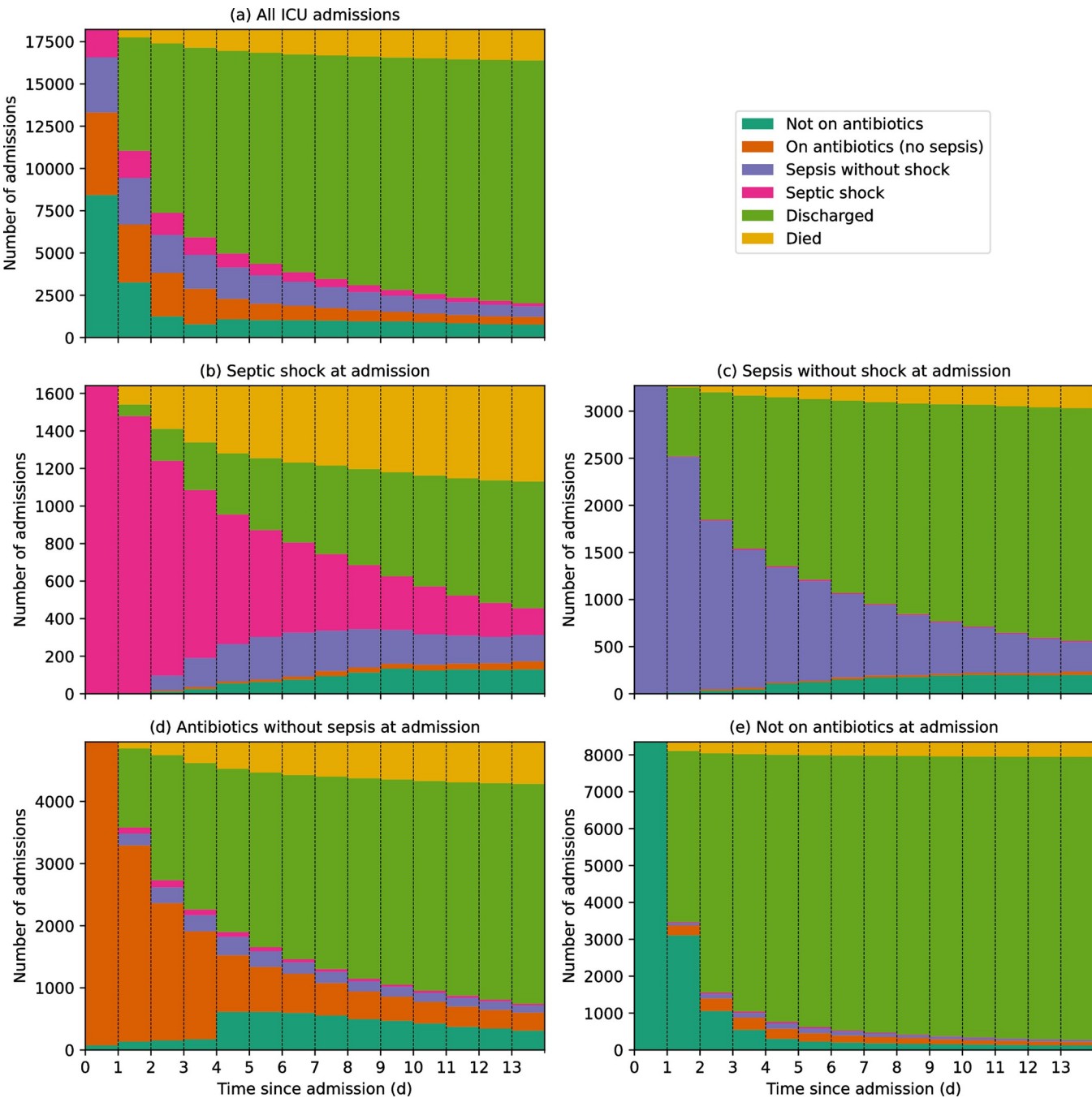

**Fig 5.** Trajectories of ICU admissions, stratified by admission sepsis status; a) All ICU admissions, b) Sepsis without shock at admission, c) Septic shock at admission, d) Antibiotics without sepsis at admission and e) Not on antibiotics at admission. We considered a patient's admission sepsis status to remain unchanged while the following conditions were met: i) a sepsis episode continued for as long as a patient's SOFA score is higher than their SOFA baseline* AND they remained on antibiotics, ii) a septic shock episode continued for as long as their lactate levels remained >2 or they remained on vasopressors. *Baseline SOFA score was defined as the lower of the two SOFA scores that contributed towards defining a sepsis episode; for all episodes of sepsis at ICU admission, baseline SOFA = 0.

Meanwhile, our analysis of antibiotic prescribing habits in the Amsterdam UMC intensive care unit found that ceftriaxone was the most commonly used antibiotic followed by vancomycin, metronidazole, ciprofloxacin and co-amoxiclav. Vancomycin was the most commonly

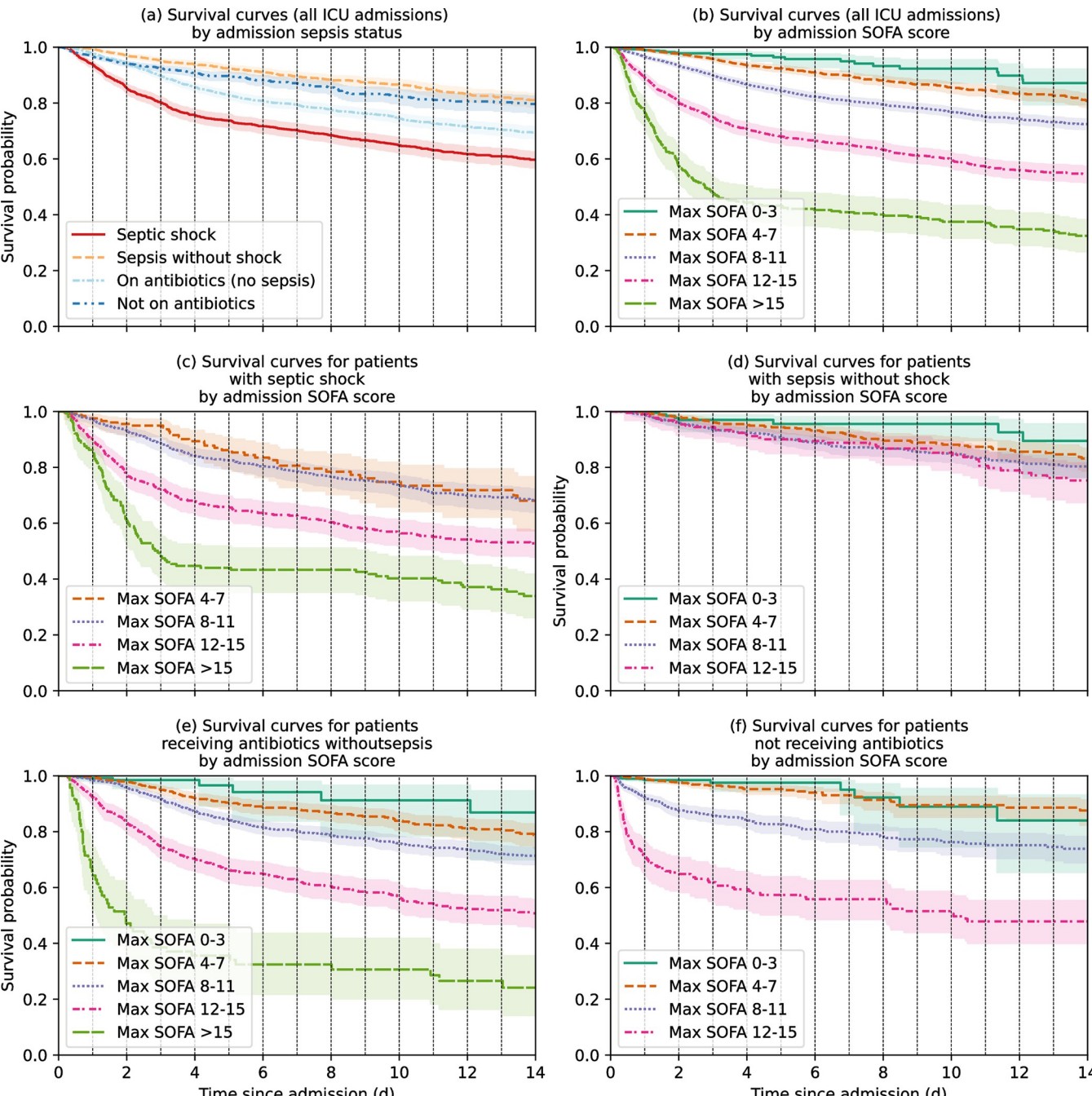

**Fig 6.** Kaplan Meier curves from cause-specific (all-mortality) Cox proportional hazard models showing survival probability based on admission sepsis status (a) and admission SOFA score (b) among all ICU admissions. 6c-f) Kaplan Meier curves from cause-specific (all-mortality) Cox proportional hazard models showing survival probability based on admission SOFA score, among patients with (c) septic shock*, (d) sepsis without shock, (e) antibiotics without sepsis and (f) no antibiotics on ICU admission. *Due to insufficient data in the '*Max SOFA 0–3*' group for patients with septic shock on admission, hazard models were calculated in comparison to the baseline hazard of the '*Max SOFA 4–7*' group; all other hazard models in 6c-f were calculated in comparison to the baseline hazard of the '*Max SOFA 0–3*' group.

used antibiotic in Rank 3 or above, with Rank 4 antibiotics rarely prescribed. Notably, Rank 3 and 4 antibiotics were used substantially less frequently (20.7% and 6.0% of antibiotic courses respectively) compared to in Shah's study, where Rank 3 and Rank 4 antibiotics made up

**Table 3. Antibiotic usage (treatment-only; excluding prophylactic\* antibiotics) in the intensive care unit.**

| Antibiotic | Total number of courses | Percentage of all courses (%) | Total number of antibiotic-days |
|---|---|---|---|
| **Rank 4:** | | | |
| Imipenem | 464 | 3.4 | 2711 |
| Meropenem | 229 | 1.7 | 1245 |
| Colistin | 108 | 0.8 | 855 |
| Amikacin | 13 | 0.1 | 55 |
| Linezolid | 4 | 0.0 | 15 |
| Neomycin sulphate | 4 | 0.0 | 12 |
| Tigecycline | 1 | 0.0 | 5 |
| Daptomycin | 0 | 0.0 | 7 |
| **Rank 3:** | | | |
| Vancomycin | 1767 | 13.0 | 6381 |
| Gentamicin | 521 | 3.8 | 1014 |
| Ceftazidime | 275 | 2.0 | 1283 |
| Piperacillin | 263 | 1.9 | 1152 |
| **Rank 2:** | | | |
| Ceftriaxone | 3696 | 27.1 | 15684 |
| Ciprofloxacin | 1057 | 7.7 | 4704 |
| Co-Amoxiclav | 844 | 6.2 | 2806 |
| Levofloxacin | 490 | 3.6 | 1979 |
| Erythromycin | 383 | 2.8 | 1499 |
| Cefuroxime | 295 | 2.2 | 414 |
| Clindamycin | 198 | 1.5 | 823 |
| Cefotaxime | 34 | 0.3 | 488 |
| Clarithromycin | 20 | 0.2 | 82 |
| Azithromycin | 19 | 0.1 | 52 |
| Norfloxacin | 5 | 0.0 | 7 |
| Moxifloxacin | 4 | 0.0 | 48 |
| **Rank 1:** | | | |
| Metronidazole | 1326 | 9.7 | 7220 |
| Co-Trimoxazole | 627 | 4.6 | 3302 |
| Amoxicillin | 485 | 3.6 | 2263 |
| Tobramycin | 295 | 2.2 | 1100 |
| Benzylpenicillin | 174 | 1.3 | 957 |
| Doxycycline | 37 | 0.3 | 141 |
| Feneticillin | 7 | 0.1 | 23 |
| Nitrofurantoin | 5 | 0.0 | 14 |
| Total | 13650 | 100 | 58341 |

\*Prophylactic antibiotics were defined as cefotaxime prescribed within 4 days of ICU admission (as part of Amsterdam UMC's Selective Digestive Decontamination regimen), all cefazoline administration, antibiotics prescribed within 24 hrs of admission for elective surgery patients, vancomycin administration any day following cardiac surgery, and all low dose (250 mg, 4 times daily) erythromycin administration.

31.9% and 11.1% of antibiotic courses. This may simply reflect differences in antibiotic resistance, and consequent microbiology guidance, based on location but warrants further investigation to ensure optimal antimicrobial stewardship is attained [21].

Lastly, antibiotic duration has also been suggested as one potential method of retrospectively identifying patients with true sepsis compared to those in whom antibiotics are started

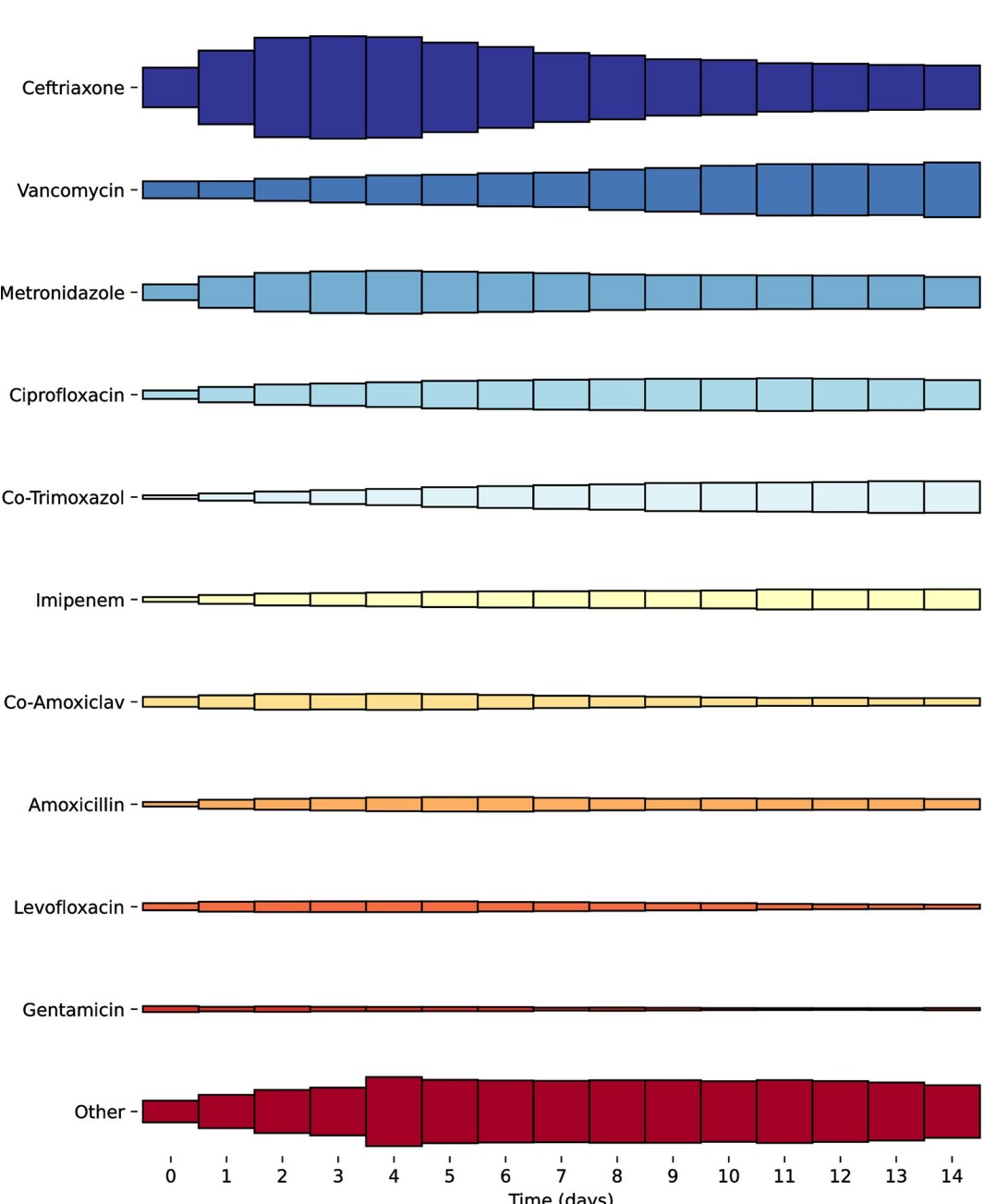

**Fig 7. Comparison of relative antibiotic use (excluding prophylactic antibiotics) among all patients in the Amsterdam UMC intensive care unit.**

for suspected infection but discontinued shortly after. We found that patients admitted with sepsis who were given 4 days or more of antibiotics had greater ICU mortality (35.8%) than those in which antibiotics were discontinued within 4 days (24.2%), though our study was limited by a lack of antibiotic data for patients discharged from ICU before completing their antibiotic course. Similar ICU mortality rates have been reported elsewhere [12].

## Limitations

Our study has several limitations. First, there was limited data available in the Amsterdam UMCdb for patients before and after their ICU stay. The absence of post-ICU mortality data prevented the calculation of in-hospital mortality and other longer-term post-ICU mortality rates. It is possible that the deaths of some ICU patients, in whom active treatment was withdrawn, were missed if they were transferred to the ward or other palliative settings prior to death. Similarly, the lack of data on cultures taken pre-ICU may underestimate the prevalence of suspected infection (and consequently sepsis) in our sensitivity analysis as discussed previously. Second, non-random missingness of data may have influenced calculation of SOFA scores and consequently alter the epidemiology of sepsis calculated. For instance, closer inspection of the daily change in component SOFA score (Figs 2–4) reveal an increasing proportion of patients where the SOFA component score could not be calculated due to missing data as time progressed. It is possible that, had SOFA scores been prospectively measured for each patient, the total SOFA score would be higher and therefore the threshold of SOFA increase $\geq 2$ met more frequently. Third, due to the retrospective nature of this study, it is not possible to determine the specific rationale for initiation of antibiotic treatment among patients with suspected sepsis, who may be started empirically on antibiotics which are stopped shortly after initiation as other differential diagnoses are confirmed. However, in our sensitivity analysis excluding patients whose new or escalated antibiotic regimen was stopped the day after initiation, only a minority of patients had their sepsis status downgraded. Finally, the ICU in this study was from an academic tertiary medical center covering a variety of specialties, including cardiothoracic surgery. This is reflected by differences in the ICU admission rate for elective surgical patients compared to other centers and consequently our results may not be representative of all ICU cohorts within the Netherlands or Europe more widely [12].

## Supporting information

**S1 Fig.**
(TIF)

**S1 File. Contains Supplementary S1–S8 Tables and S1 Fig.** This includes information about MCU admissions, results from sensitivity analyses, and Cox proportional hazards ratios.
(DOCX)

## Acknowledgments

A CC BY or equivalent licence is applied to the AAM arising from this submission. We would like to thank Professor Stephen J Eglen for supervision.

## Author Contributions

**Formal analysis:** Christopher Y. K. Williams, Tom Edinburgh.

**Investigation:** Tom Edinburgh.

**Methodology:** Tom Edinburgh.

**Supervision:** Paul W. G. Elbers, Patrick J. Thoral, Ari Ercole.

**Validation:** Christopher Y. K. Williams.

**Writing – original draft:** Christopher Y. K. Williams, Tom Edinburgh.

**Writing – review & editing:** Christopher Y. K. Williams, Tom Edinburgh, Paul W. G. Elbers, Patrick J. Thoral, Ari Ercole.

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
