## [Decision Letter · Decision Letter 0]

4 Jan 2024

PONE-D-23-32136Application of the Sepsis-3 criteria to describe sepsis epidemiology in the Amsterdam UMCdb intensive care datasetPLOS ONE

Dear Dr. Edinburgh,

Thank you for submitting your manuscript to PLOS ONE. After careful consideration, we feel that it has merit but does not fully meet PLOS ONE’s publication criteria as it currently stands. Therefore, we invite you to submit a revised version of the manuscript that addresses the points raised during the review process.

We look forward to receiving your revised manuscript.

Kind regards,

Dong Wook Jekarl

Academic Editor

PLOS ONE

Journal Requirements:

Did you know that depositing data in a repository is associated with up to a 25% citation advantage (https://doi.org/10.1371/journal.pone.0230416)? If you’ve not already done so, consider depositing your raw data in a repository to ensure your work is read, appreciated and cited by the largest possible audience. You’ll also earn an Accessible Data icon on your published paper if you deposit your data in any participating repository (https://plos.org/open-science/open-data/#accessible-data).

"TE is funded by Engineering and Physical Sciences Research Council (EPSRC) National Productivity

Investment Fund (NPIF) EP/S515334/1, reference 2089662. A CC BY or equivalent licence is applied

to the AAM arising from this submission. We would like to thank Professor Stephen J Eglen for his

supervision and mentorship of TE’s PhD."

Please be informed that funding information should not appear in the Acknowledgments section or other areas of your manuscript. We will only publish funding information present in the Funding Statement section of the online submission form. 

"TE is funded by Engineering and Physical Sciences Research Council (EPSRC) National Productivity Investment Fund (NPIF) EP/S515334/1, reference 2089662."

"TE is funded by Engineering and Physical Sciences Research Council (EPSRC) National Productivity Investment Fund (NPIF) EP/S515334/1, reference 2089662."

5. We note that you have indicated that there are restrictions to data sharing for this study. For studies involving human research participant data or other sensitive data, we encourage authors to share de-identified or anonymized data. However, when data cannot be publicly shared for ethical reasons, we allow authors to make their data sets available upon request. For information on unacceptable data access restrictions, please see http://journals.plos.org/plosone/s/data-availability#loc-unacceptable-data-access-restrictions. 

Reviewers' comments:

Reviewer's Responses to Questions

**Comments to the Author**

1. Is the manuscript technically sound, and do the data support the conclusions?

Reviewer #1: Yes

Reviewer #2: Yes

Reviewer #3: Yes

Reviewer #4: Yes

2. Has the statistical analysis been performed appropriately and rigorously? 

Reviewer #1: No

Reviewer #2: Yes

Reviewer #3: Yes

Reviewer #4: Yes

3. Have the authors made all data underlying the findings in their manuscript fully available?

Reviewer #1: Yes

Reviewer #2: Yes

Reviewer #3: Yes

Reviewer #4: Yes

4. Is the manuscript presented in an intelligible fashion and written in standard English?

Reviewer #1: Yes

Reviewer #2: No

Reviewer #3: Yes

Reviewer #4: Yes

5. Review Comments to the Author

Reviewer #1: PONE-D-23-32136

Williams et al present an analysis of the epidemiology and sepsis in patients in a mixed ICU in the Netherlands using the open access Amsterdam UMCdb. Overall they have several interesting findings, including that a significant proportion of patients progressed from sepsis to septic shock and a relatively lower mortality associated with sepsis than non-infectious conditions. The results are presented clearly, the figures are clear and support the manuscript, and the discussion in appropriate. My main concern (as highlighted below) is on the comparison between patients who received less vs greater than 4 days of antibiotics (this led to me marking “No” under “Has the statistical analysis been performed appropriately and rigorously?”).

COMMENTS

1. Methods, Identification of sepsis and septic shock – how did you define “a new course of antibiotics”? Could this have been just for one or two days? The concern would be including sick patients who get empiric antibiotics but then the providers quickly realize that they have a non-infectious illness.

2. Methods – were you able to gather data for patients after they left the ICU, or just while in the ICU? After reading the discussion I realize the answer but this should be clear in the methods and is important when considering duration of antibiotic therapy (among other variables).

3. Results, identification of sepsis and septic shock – how did the authors define when an episode “ended” (and therefore when the patient would be eligible for a new episode)?

4. Results, identification of sepsis and septic shock – For the sensitivity analysis requiring 6+ hours of NA infusion, what was the difference that was statistically significant? This is confusing as written and I am unsure what the comparison is here.

5. Results, identification of sepsis and septic shock – the last sentence of this section reports shorter LOS among patients who received < 4 days antibiotics than patients who received > 4 days antibiotics. However, if you don’t capture antibiotics that were prescribed after patients left this ICU, then patients probably got shorter antibiotic durations because they were discharged (instead of being discharged once they stopped antibiotics and where therefore “better”). Please clarify.

6. Results, ICU mortality – the discussion here about <4 vs >4 days antibiotics is confusing for similar reasons as above.

7. Table 1. I am a little surprised that only 57.9% and 42.0% of patients with sepsis and septic shock, respectively, received 4 days of antibiotics. This gets back to comment #1 – did these patients receive short antibiotic courses? I would assume that patients who actually had sepsis or septic shock would receive > 5-7 days of antibiotics (unless they died or were discharged)

8. Figure 6 – recommend re-labeling the titles for panels c-f to be “Survival curves among patients with…” to make these panels clearer and harmonize with panels a and b.

9. Figure S1 – at what timepoint did you define the outcome? Ie, how long did you follow patients to determine whether they were still in the ICU?

10. Table S2 – as above, unless I am confused about which patients were eligible to receive < 4 vs >4 days of antibiotics, then I am not sure this is an appropriate comparison.

Reviewer #2: Generally is a very interesting manuscript. However:

English language editing is needed.

Legends with explanations under ever figure are needed

it would be very interesting if authors added the causes of death in both subgroups (> 4 days and < 4 days.)

Reviewer #3: Despite the fact that this is a retrospective cohort study, with limited information prior to and after ICU admission, it is a remarkable study, giving unique information on sepsis epidemiology according to Sepsis-3 definition. Well-wrighten, well-designed study and great depiction of the results. Well-structured discussion with appropriate focus on study insights and previous investigation. The inclusion of Ethics Approval number for secondary research and statement of harmonization with the Helsinki Declaration would be appreciated as informed consent is waived.

Reviewer #4: The manuscript is well-written and thorough. The authors have provided comprehensive and detailed data and offered information on various epidemiological aspects of septic patients in the ICU using Sepsis-3 criteria. The research article features an advanced-level statistical analysis that ensures high quality. The conclusions are supported by the data and cover essential aspects for clinicians. I appreciate that you mentioned the disparity in the frequency of particular antibiotics among various ICUs due to the varying resistance profiles.

The infection was defined as a new course of antibiotics or an escalation in antibiotic therapy. In the primary analysis, the performance of microbial cultures was not considered in the diagnosis of infection. How did the clinicians decide to initiate the antibiotic treatment if not based on cultures? It will be interesting to know if they used some biomarkers (CRP, procalcitonin).

In the methods section, it may be helpful to mention the measurement of survival probability using Kaplan-Meier curves, as this is already described in the figures section.

I would like to ask for clarification on the exclusion criteria. Between paragraphs one and four of the methods, there is a difference in the day of admission when the SOFA score was measured. Is day 0 set as the first day of admission?

If the Amsterdam ICU’s protocols are published somewhere it would be interesting to be cited.

I would like to share a few minor comments. I would suggest that the percentages have the same form: numbers or letters. For example, paragraph three of the Results section: “Seventy-three percent (2,423/3,304) of ICU admissions for sepsis without shock were associated with IV antibiotics for at least 4 days or until end of ICU stay, compared to 84.0% (1,389/1,654) admissions with septic shock.”, or paragraph 3 of the abstract: “Forty-eight percent of emergency medical admissions and 37.0% of emergency surgical admissions were for sepsis.”

Some additional corrections: We sought to apply the Sepsis-3 criteria "to" characterise the septic cohort in the Amsterdam University Medical Centres database (Amsterdam UMCdb).

Because pre-ICU data "are" limited in the Amsterdam UMC database.

6. PLOS authors have the option to publish the peer review history of their article (what does this mean?). If published, this will include your full peer review and any attached files.

Reviewer #1: **Yes: **Max Adelman

Reviewer #2: **Yes: **Maria Lagadinou

Reviewer #3: **Yes: **Michailides Christos

Reviewer #4: **Yes: **Charikleia Chourpiliadi

---

## [Author Response · Author response to Decision Letter 0]

16 Apr 2024

Our full response to the reviewers has been attached as a separate file, but I have copied the text here too.

Reviewer #1: PONE-D-23-32136

  Williams et al present an analysis of the epidemiology and sepsis in patients in a mixed ICU in the Netherlands using the open access Amsterdam UMCdb. Overall they have several interesting findings, including that a significant proportion of patients progressed from sepsis to septic shock and a relatively lower mortality associated with sepsis than non-infectious conditions. The results are presented clearly, the figures are clear and support the manuscript, and the discussion in appropriate.

Response: We thank the reviewer for their positive feedback.

My main concern (as highlighted below) is on the comparison between patients who received less vs greater than 4 days of antibiotics (this led to me marking "No" under "Has the statistical analysis been performed appropriately and rigorously?").

Response: We thank the reviewer for raising this concern. Please see our full response below.

  COMMENTS

  1. Methods, Identification of sepsis and septic shock – how did you define "a new course of antibiotics"? Could this have been just for one or two days? The concern would be including sick patients who get empiric antibiotics but then the providers quickly realize that they have a non-infectious illness.

Response: We have followed the approach adopted by Shah et al (https://www.ncbi.nlm.nih.gov/pmc/articles/PMC8508729/) to allow consistent comparison between datasets. We defined ‘a new course of antibiotics’ as ‘one or more doses of antibiotics prescribed for a patient not already receiving antibiotics’ and have clarified the definition in the Methods (lines 96-97) accordingly. We have additionally conducted a sensitivity analysis where patients receiving a new or escalate antibiotic regimen, with associated SOFA score increase, for a single day only were not classified as sepsis (lines 199-204). In this analysis, there were 160 fewer emergency ICU admissions classified as either septic shock or sepsis without shock compared to the main analysis (Table S6).

2. Methods – were you able to gather data for patients after they left the ICU, or just while in the ICU? After reading the discussion I realize the answer but this should be clear in the methods and is important when considering duration of antibiotic therapy (among other variables).

Response: Data was available from ICU admission to ICU discharge – we have clarified this in the description of the Amsterdam UMCdb dataset in the Methods (lines 86-87).

 3. Results, identification of sepsis and septic shock – how did the authors define when an episode "ended" (and therefore when the patient would be eligible for a new episode)?

Response: Patients were eligible for a new sepsis episode if they met the criteria for sepsis more than 72 hours after a previous sepsis episode. We have updated the Methods to reflect this (lines 116-118).

4. Results, identification of sepsis and septic shock – For the sensitivity analysis requiring 6+ hours of NA infusion, what was the difference that was statistically significant? This is confusing as written and I am unsure what the comparison is here.

Response: The statistically significant difference is the re-classification (in the sensitivity analysis requiring >6 hours of noradrenaline infusion) of 100 fewer emergency ICU admissions classified as septic shock compared to the main analysis. We have clarified this text in the Results (lines 192-193).

 5. Results, identification of sepsis and septic shock – the last sentence of this section reports shorter LOS among patients who received < 4 days antibiotics than patients who received > 4 days antibiotics. However, if you don't capture antibiotics that were prescribed after patients left this ICU, then patients probably got shorter antibiotic durations because they were discharged (instead of being discharged once they stopped antibiotics and where therefore "better"). Please clarify.

Response: We thank the reviewer for pointing this out – we initially sought to follow the same columns presented by Shah et al, but we agree this is confusing. We have updated Table 2 to include separate columns, denoting whether patients who had <4 days of antibiotics were discharged from the ICU within 4 days ‘Antibiotics until discharge (ICU length of stay <4 days)’ or if they had their antibiotics stopped within 4 days ‘Antibiotics <4 days (ICU length of stay ≥4 days)’. 

6. Results, ICU mortality – the discussion here about <4 vs >4 days antibiotics is confusing for similar reasons as above.

Response: Similar to the above comment, our updated columns in Table 2 clarify the differences in ICU mortality depending on whether they were discharged, with 0% ICU mortality among patients discharged from the ICU in <4 days (by definition), 24.2% mortality among those who received antibiotics for <4 days and had an ICU length of stay ≥4 days, and 35.8% among those receiving antibiotics for at least 4 days or until death.

7. Table 1. I am a little surprised that only 57.9% and 42.0% of patients with sepsis and septic shock, respectively, received 4 days of antibiotics. This gets back to comment #1 – did these patients receive short antibiotic courses? I would assume that patients who actually had sepsis or septic shock would receive > 5-7 days of antibiotics (unless they died or were discharged)

Response: 

We thank the reviewer for this comment. We agree that the way this was initially presented was confusing – the row ‘IV antibiotics for first 4d (*), n (%)’ represented the number of patients receiving antibiotics consecutively for the first 4 days AND had admission length of stay >= 4 days, which was a lower number of patients (57.9% and 42.0% of patients with sepsis and septic shock) compared to the row below, ‘IV antibiotics for at least 4d (†), n (%)’ which included all patients who received antibiotics for at least 4 days in total OR until ICU death/discharge.

We have subsequently revised these rows, excluding the former ‘IV antibiotics for first 4d (*), n (%)’ row and reporting only the latter row, which shows that 87.9% and 89.0% of patients with septic shock and sepsis, respectively, received antibiotics for at least 4 days in total or until ICU death/discharge.

8. Figure 6 – recommend re-labeling the titles for panels c-f to be "Survival curves among patients with..." to make these panels clearer and harmonize with panels a and b.

Response: We have updated the Figure 6 titles to harmonise panels c-f with a and b as requested.

9. Figure S1 – at what timepoint did you define the outcome? Ie, how long did you follow patients to determine whether they were still in the ICU?

Response: We thank the reviewer for this comment – we have added ‘(measured 15 days after ICU admission)’ to clarify the 15 day cut-off timepoint used to define the outcome.

10. Table S2 – as above, unless I am confused about which patients were eligible to receive < 4 vs >4 days of antibiotics, then I am not sure this is an appropriate comparison.

Response: As described above for Table 2, we have updated Table S2 to use the same format as Table 2, denoting whether patients who had <4 days of antibiotics were discharged from the ICU within 4 days ‘Antibiotics until discharge (ICU length of stay <4 days)’ or if they had their antibiotics stopped within 4 days ‘Antibiotics <4 days (ICU length of stay ≥4 days)’.

Reviewer #2: Generally is a very interesting manuscript. However:  English language editing is needed.  Legends with explanations under ever figure are needed  . It would be very interesting if authors added the causes of death in both subgroups (> 4 days and < 4 days.)

Response: We thank the reviewer for their comments. We note the other reviewers have all highlighted that the manuscript was well-written. We kindly invite this reviewer to provide further details of which parts of the manuscript they believe require English language editing if they still have concerns. 

In addition, please note that full Figure legends are already provided on page 20 of the manuscript.

Unfortunately, we do not have data available on the causes of death of patients in the Amsterdam UMCdb.

Reviewer #3: Despite the fact that this is a retrospective cohort study, with limited information prior to and after ICU admission, it is a remarkable study, giving unique information on sepsis epidemiology according to Sepsis-3 definition. Well-wrighten, well-designed study and great depiction of the results. Well-structured discussion with appropriate focus on study insights and previous investigation. The inclusion of Ethics Approval number for secondary research and statement of harmonization with the Helsinki Declaration would be appreciated as informed consent is waived.

Response: We thank the reviewer for their positive feedback.

Please note that formal ethics approval was not required for this secondary research since Amsterdam UMCdb has already received ethical approval that covers secondary research using the de-identified data. The training courses required for data access were completed prior to access and analysis of the Amsterdam UMCdb dataset.

Reviewer #4: The manuscript is well-written and thorough. The authors have provided comprehensive and detailed data and offered information on various epidemiological aspects of septic patients in the ICU using Sepsis-3 criteria. The research article features an advanced-level statistical analysis that ensures high quality. The conclusions are supported by the data and cover essential aspects for clinicians. I appreciate that you mentioned the disparity in the frequency of particular antibiotics among various ICUs due to the varying resistance profiles.

Response: We thank the reviewer for their positive feedback.

The infection was defined as a new course of antibiotics or an escalation in antibiotic therapy. In the primary analysis, the performance of microbial cultures was not considered in the diagnosis of infection. How did the clinicians decide to initiate the antibiotic treatment if not based on cultures? It will be interesting to know if they used some biomarkers (CRP, procalcitonin).

Response: Due to the nature of this retrospective, observational study, it is not possible to determine the specific rationale for initiation of antibiotic treatment among patients with suspected sepsis, who may be started empirically on antibiotics which are stopped shortly after initiation as other differential diagnoses are confirmed. However, in our sensitivity analysis excluding patients whose new or escalated antibiotic regimen was stopped the day after initiation (Table S6; conducted in response to Reviewer #1’s comment, above), only a minority of patients had their sepsis status downgraded. We have added this into the Limitations section of the Discussion (lines 415-422).

In the methods section, it may be helpful to mention the measurement of survival probability using Kaplan-Meier curves, as this is already described in the figures section.

Response: We have added this clarification to the Statistical analysis section of the Methods.

I would like to ask for clarification on the exclusion criteria. Between paragraphs one and four of the methods, there is a difference in the day of admission when the SOFA score was measured. Is day 0 set as the first day of admission?

Response: We thank the reviewer for spotting this and have amended the text to set day 0 as the first day of admission.

If the Amsterdam ICU's protocols are published somewhere it would be interesting to be cited.  I would like to share a few minor comments. I would suggest that the percentages have the same form: numbers or letters. For example, paragraph three of the Results section: "Seventy-three percent (2,423/3,304) of ICU admissions for sepsis without shock were associated with IV antibiotics for at least 4 days or until end of ICU stay, compared to 84.0% (1,389/1,654) admissions with septic shock.", or paragraph 3 of the abstract: "Forty-eight percent of emergency medical admissions and 37.0% of emergency surgical admissions were for sepsis."

Response: We have followed standard grammatical practice in avoiding the use of numbers at the start of a sentence. We are happy to defer to the recommendations of the Editor and amend the use of number vs letters if requested to do so.

Some additional corrections: We sought to apply the Sepsis-3 criteria "to" characterise the septic cohort in the Amsterdam University Medical Centres database (Amsterdam UMCdb). Because pre-ICU data "are" limited in the Amsterdam UMC database.

Response: We thank the reviewer for spotting these typos and have amended them accordingly.

---

## [Editor Report · Decision Letter 1]

7 May 2024

Application of the Sepsis-3 criteria to describe sepsis epidemiology in the Amsterdam UMCdb intensive care dataset

PONE-D-23-32136R1

Dear Dr. Edinburgh,

We’re pleased to inform you that your manuscript has been judged scientifically suitable for publication and will be formally accepted for publication once it meets all outstanding technical requirements.

Kind regards,

Dong Wook Jekarl

Academic Editor

PLOS ONE
---

## [Editor Report · Acceptance letter]

13 May 2024

PONE-D-23-32136R1 

PLOS ONE

Dear Dr. Edinburgh, 

I'm pleased to inform you that your manuscript has been deemed suitable for publication in PLOS ONE. Congratulations! Your manuscript is now being handed over to our production team.

Kind regards, 

on behalf of

Dr. Dong Wook Jekarl 

Academic Editor

PLOS ONE